# The Effects of High-Altitude Mountaineering on Cognitive Function in Mountaineers: A Meta-Analysis

**DOI:** 10.3390/ijerph20065101

**Published:** 2023-03-14

**Authors:** Lun Li, Yun Zhou, Shisi Zou, Yongtai Wang

**Affiliations:** 1College of Physical Education, China University of Geosciences (Wuhan), Wuhan 430074, China; lilun@cug.edu.cn (L.L.);; 2College of Health Sciences and Technology, Rochester Institute of Technology, Rochester, NY 14623, USA

**Keywords:** high-altitude, mountaineering, mountaineer, cognitive function, meta-analysis

## Abstract

Background: Nowadays, with the convenience of international traveling and driven by many individuals’ fond dreams of challenging high-altitude exercises, high-altitude mountaineering is becoming increasingly popular worldwide. Therefore, we performed a meta-analysis to determine the effects of high-altitude mountaineering on cognitive functions in mountaineers before and after climbing. Methods: After a thorough electronic literature search and selection, eight studies were included in this meta-analysis, and the conducted test cycles ranged from 8 to 140 days. Eight variables were included in this meta-analysis: the Trail-Making Test (TMB), Digit Span-Forward (DSF), Digit Span-Backward (DSB), Finger Tapping Test-Right (FTR) Finger Tapping Test-Left (FTL), Wechsler Memory Scale Visual (WMSV), the Aphasia Screening Test (Verbal Items) (AST-Ver), and the Aphasia Screening Test (Visual Motor Errors) (AST-Vis). The effect sizes (ES) and forest plots of these eight variables were generated. Results: Five variables (TMB, ES = 0.39; DSF, ES = 0.57; FTR, ES = 0.50; FTL, ES = 0.16; WMSV, ES = 0.63) out of eight were significantly improved after high-altitude mountaineering, whereas the ES values of DSB, AST-Ver, and AST-Vis did not show significant improvement after climbing. Conclusion: Despite two limitations, namely, methodological issues inherent in the meta-analysis and the inability to explain high heterogeneity between studies, this study is the first meta-analysis that has attempted to specify and compare the cognitive functions of mountaineers before and after high-altitude mountaineering. Furthermore, as a short-term plateau exercise, high-altitude mountaineering has no significant negative impacts on the cognitive functions of climbers. Future research is needed for a long period of high-altitude mountaineering.

## 1. Introduction

High-altitude mountaineering can bring about a sense of performance achievement and heightened willpower to challenge one’s environment [1]. Currently, with the convenience of international travel, high-altitude exercises, such as hiking, mountaineering, and skiing, are no longer just fond dreams for many individuals. High-altitude climbing [2] refers to the act of climbing elevations with altitudes exceeding 3500 m.

Studies have shown that hypobaric and anoxic environments in high-altitude areas affect people’s cognitive functions, such as short-term memory, attention span, attention conversion ability, and thinking and judgment ability [3,4,5,6,7], which could be potentially hazardous to individual health, both physically and mentally [8,9]. Such conditions can lead to irrational decisions, leading to falls, frostbite, accidents, fatigue, and death.

However, only a few studies have explored the physiological and cognitive changes that occur during high-altitude exploration [10,11]. In addition, most high-altitude mountaineering studies on cognitive functioning are of small sample size, and the effects of high-altitude mountaineering on cognitive function have not been well documented in a systematical format. In this meta-analysis, the effects of altitude exposure on the cognitive function of mountaineers were systematically reviewed, and applied neuropsychological tests were classified according to their superior cognitive domains. Therefore, the purpose of this meta-analysis was to critically assess the effects of high-altitude mountaineering exercise on cognitive function in terms of executive function, motor speed, memory function, and verbal function in mountaineers. This is the first review to study the impact of high-altitude mountaineering on the cognitive function of mountaineers by means of meta-analysis with each neuropsychological test assigned to its cognitive field. Hopefully, the results of this study may provide valuable references for further improving the protective measures of climbers’ cognitive functions under plateau environments, assessing the cognitive functions of the nervous systems of climbers, and developing new training measures for climbers, thereby ensuring the safety of climbers and the healthy development of mountaineering worldwide.

## 2. Methods

### 2.1. Search Strategy

A computer information retrieval system was used to obtain data from China and abroad, including searches from the China National Knowledge Infrastructure (CNKI), WanFang Database, PubMed, Web of Science, Google Scholar, and Baidu Scholar. The timeframe for the information retrieval was limited between January 1980 and September 2022 with the terms “high-altitude”, “mountaineering”, and “cognitive function”. Theme words and keywords were combined to retrieve data, and the retrieved literature was further checked by another researcher.

All the following seven required conditions should be met in the eligible research criteria. (1) Research type: scientific research; (2) research subject: the subjects live at low altitude all year round, climb mountains at high altitude, and are healthy (aged 18–60 years); (3) research method: independent literature with similar hypotheses and research methods; (4) original data with complete information; (5) the experimental site is at or above an ultra-high-altitude environment (the altitude is more than 3500 m), and the exposure time is more than 1 week; (6) academic papers in either Chinese or English; and (7) all the data needed for the meta-analysis were derived from the means and standard deviations of the pre-experiments and post-experiments. For overlapping publications of the same researchers, the most complete and high-quality studies were selected.

### 2.2. Study Selection

In line with the literature search strategy, 597 articles were first selected among those published between January 1980 and September 2022. After the second round of review, 69 duplicate articles were eliminated, and 528 articles were highlighted in the second round of selection. After carefully reading each article title and abstract, 437 not-relevant articles and 65 articles were reviewed, meta or case reports, and 26 articles which met our standards were obtained after excluding these other articles. In total, 18 articles were eliminated because 13 articles took plateau residents and stationed army soldiers as research subjects, 3 discussed situations below 3600 m in altitude, 1 had incomplete data, and another study used repetitive data. Finally, only 8 articles were included in the study [12,13,14,15,16,17,18,19], among which, 1 was published in Chinese and 7 were in English with one hybrid experiment. The experimental group and the control group within the same test were analyzed. Hence, two independent sets of data from the study of Hornbein [12] were introduced into the meta-analysis. A systematic review and meta-analysis were conducted as per the Preferred Reporting Items for Systematic Reviews and Meta-Analyses (PRISMA) guidelines [20]. The flow chart of the literature screening is shown in Figure 1.

### 2.3. Data Extraction

General outlooks, including the researchers, publication time, research types, frequency, and the duration of intervention studies were all analyzed using Excel in Microsoft Office 2021. Five sets of specific data from the study of Philip [16] were processed by using SAS 9.1.3 (SAS Institute Inc., Cary, NC, USA) for the means and the standard deviations of the related tests. The Finger Rapping Test in Hornbein’s experiment was seen as the Finger Tapping Test to be combined with other research data. As shown in Table 1, all eight articles covered the eight testing indexes from the four domains of cognitive functions.

Among the retrieved studies, 8 out of 26 were selected based on the literature-inclusive criteria, with 173 research subjects (147 males and 26 females) aged between 18 and 60 years old for a test cycle of all included studies lasting from 8 to 140 days. The details are shown in Table 2.

### 2.4. Statistical Analysis

To ensure an effective meta-analysis, at least three sets of statistics from each group should be included; namely, statistics from the pre-test and post-test are requisite. The test consisted of the Trail-Making Test Part B (TMB), Finger Tapping Test-Left (FTL), Finger Tapping Test-Right (FTR), Digit Span Test-Forward (DSF), Digit Span Test-Backward (DSB), Wechsler Memory Scale Visual (WMSV), Aphasia Screening Test-Visual Motor Errors (AST-Vis), and Aphasia Screening Test-Verbal Items (AST-Ver), which are reflective of the four domains of cognitive functions, namely, executive function, motor speed, memory function, and verbal function. The effect size (ES) of each test was computed [21]. Usually, researchers define ES as small (0.2 ≤ ES < 0.4), medium or moderate (0.4 ≤ ES < 0.6), and large (ES ≥ 0.6) with *p* < 0.05 [22]. (Cohen, 1962) The analyses of overall ES were implemented using a random effects model [23].

## 3. Results

The ES, forest plots, and the results of the above-mentioned eight test variables are presented in the following tables.

### 3.1. TMB

Three studies in which 56 participants were reported to have the test on TMB were analyzed, and three sets of statistics were generated from the comparison tests of the same groups. As shown in Table 3, there was no significant difference between the pre-test and the post-test on the participants’ TMB (95% CI, −0.05 to 0.83), and the mean ES was 0.39, *p* > 0.05.

### 3.2. DSF

Six studies with 123 participants tested on DSF were analyzed, and six sets of statistics were generated from the comparison tests of the same groups. As shown in Table 4, there was a significant difference regarding DSF (95% CI, 0.24–0.90) between the pre-test and the post-test, and the mean ES was 0.57, *p* < 0.01.

### 3.3. DSB

Three sets of statistics were generated from the comparison tests of the same groups, with three articles involving 35 participants tested on DSB. As suggested in Table 5, there was no significant difference in DSB (95% CI, −0.83 to 0.30) between the pre-test and the post-test, and the mean ES was −0.26, *p* > 0.05.

### 3.4. FTL

According to four sets of statistics from three studies with two inter-group and two inner-group comparison test designs, 58 participants had FTL and revealed no significant difference between the pre-test and the post-test in this respect (95% CI, −0.27 to 0.59), and the mean ES was 0.16, *p* > 0.05 (Table 6).

### 3.5. FTR

Table 7 shows the findings of four sets of statistics from three studies on FTR. Altogether, 58 participants took the test with two inter-group and two inner-group comparison test designs. The results suggested a significant difference between the pre-test and the post-test in this respect (95% CI, 0.06–0.94), and the mean ES was 0.50, *p* < 0.05.

### 3.6. WMSV

The findings from the WMSV participated in by 49 test-takers, as reported in two studies, indicated a significant difference between the pre-test and the post-test (95% CI, 0.14–1.12), with the mean ES being 0.63, *p* < 0.05 (Table 8). In this category, two studies with a within-subject design and one study with a between-subject design contributed to the three sets of statistics from which the results were generated.

### 3.7. AST-Ver

For AST-Ver, three sets of statistics from two studies with a within-subject design and one with a between-subject design reported 49 participants’ behaviors, and there was no significant difference between the pre-test and the post-test (95% CI, −0.38 to 0.13). The mean ES was −0.35, *p* > 0.05 (Table 9).

### 3.8. AST-Vis

Two studies reported 63 participants in the Aphasia Test (Visual Motor Errors), from which, three sets of statistics with two within-subject designs and one between-subject design generated the finding, as shown in Table 10. There was no significant difference between the pre-test and the post-test (95% CI, −0.43 to 0.42), and the mean ES was 0, *p* > 0.05.

## 4. Discussion

Compared with many other tissues in the body, the brain is more dependent on a constant oxygen supply, so some brain regions are particularly vulnerable to hypoxia. To some extent, due to their distal position in the distribution of blood vessels [24], this region includes the hippocampus, basal ganglia, and cerebral cortex [25,26]. 

According to the ES from the selected literature, the findings from this meta-analysis suggested that high-altitude mountaineering had no impact on mountaineers’ executive and verbal functions, whereas certain impacts on motor speed and memory function were observed.

### 4.1. Executive Function

“Executive function”, also known as “executive function and cognitive control”, is a series of cognitive processes essential for cognitive control, namely, the conduct to opt for and successfully monitor those helpful for obtaining one’s goals [27]. In the literature, TMB is widely regarded as a suitable test of executive function [28].

The results of the meta-analysis suggested that mountaineers showed no significant cognitive decline in TMB after high-altitude mountaineering, as the TMB ES of 0.39 represented a very small influence. The hippocampus plays an important role in learning and memorizing, as does the cerebral cortex, which is also responsible for the advanced processes of the human brain, such as executive function [24,29]. In comparison, the report of Charles and Gregory showed that there was no numerical influence on TMB, which is different from our finding that high-altitude mountaineering had no significant impact on mountaineers’ executive function. The underlying reason for this might be related to the length of duration in high-altitude areas. Hence, future research calls for considering the cycle time of high-altitude mountaineering on executive function, such as the study on TMB.

### 4.2. Motor Speed

Motor speed is the key predictive index of cognitive and executive function. A common measurement of motor speed is the FTS, which is typically administered as part of a neurological or neuropsychological assessment [30].

A computation of the ES of the FTL and the FTR revealed 0.16 with a small influence degree and 0.50 with a medium influence degree, respectively; however, the FTL did not reach a significant level (*p* < 0.05), whereas the FTR reached a significant level. On the one hand, Petiet interpreted that the performance of the left hand improved before and after the test, which may be due to the practice effect [17]. On the other hand, the difference between the left and right hands could be attributed to all the test-takers’ right-handedness: their left hands were comparatively less adroit in precision operation and were less influenced after high-altitude mountaineering, whereas their right-hand proficiency showed a significant decline due to plateau hypoxia [31].

### 4.3. Memory Function

Memory is the brain’s function of encoding, storing, and retrieving information. Memory is vital for experiencing in the limbic system and it will keep information as time goes by so as to affect future behavior [32]. The tests of DSF and DSB are both tests on verbal memory function [24], whereas the WMSV is the assessment tool on non-verbal memory function [33].

The ES of DSF and DSB from this study were 0.57 and −0.26, respectively, and the effect size of WMSV was 0.63. 

The authors of the two studies held the view that high altitude had a negative impact on memory function, but the impact was not necessarily absolute and depended on various conditions and the mountaineers’ emotions. Both studies lasted over five weeks, and the mountaineers were on mountains at least 7500 m in altitude. This also confirmed that prolonged exposure to high altitude, including insufficient sleep at night resulting from hypoxia, could affect mountaineers’ working memory functions [34,35].

A study at Kangqin Rongjia Base Camp (5350 m) by Pagani et al. [36] showed a reduction in memory function. However, some studies have measured the memory function of mountaineers before and after mountaineering, and the results demonstrate that the performance of working memory has not decreased, which may indicate that this long-term exposure to high altitude has no short-term or long-term negative impact on working memory [19].

### 4.4. Verbal Function

Natural language is the medium for non-domain-specific thinking, serving to integrate the outputs of a variety of domain-specific conceptual faculties (or central-cognitive “quasi-modules”) [37]. The AST is widely used for the assessment of cognitive decline and the degree of such decline in verbal cognition which is essential in language [38,39].

It was found after close computation that both AST-Ver and AST-Vis exerted a small influence with respect to the ES of −0.35 and 0; moreover, neither of them reached a significant level (*p* < 0.05), which, to some extent, suggested that high-altitude mountaineering had no negative impact on the mountaineers’ verbal function. However, Hornbein [12] believed that the insignificant impact on verbal function could be linked to the time of the testing, which took place at a certain period after mountaineering, whereas in the timely test of a mock experiment, high-altitude mountaineering had a significant impact on the mountaineers’ verbal function. 

### 4.5. Limitations

This meta-analysis has some limitations due to the limited literature and the gap between available data and studies. Meanwhile, other possible factors through which high-altitude mountaineering could impact cognition include: the exact altitude, local weather conditions, longitude, and latitude; the cold environment’s impact on the testing equipment; the high altitude’s impact on mountaineers; the disparity among mountaineers’ stamina, skills, and experience; and the frequency and extent of using auxiliary oxygen, all of which might limit the scope of this meta-analysis.

## 5. Conclusions

High-altitude mountaineering has a negative impact on mountaineers’ motor speed and memory function, and this impact is more significant in the domain of memory function, especially in language working memory. Meanwhile, it seems that high-altitude mountaineering has no negative impact on executive function and verbal function, but further studies on executive function and verbal function are needed in future research. Future studies should focus on common consensus and complement each other. For example, one possibility is to create a standardized test framework, which can make different experiments more comparable and help determine the basic influencing factors. Then, follow-up studies should develop a set of simple testing tools that can quickly and accurately test the cognitive function indicators of climbers during the climbing process.

## Figures and Tables

**Figure 1 ijerph-20-05101-f001:**
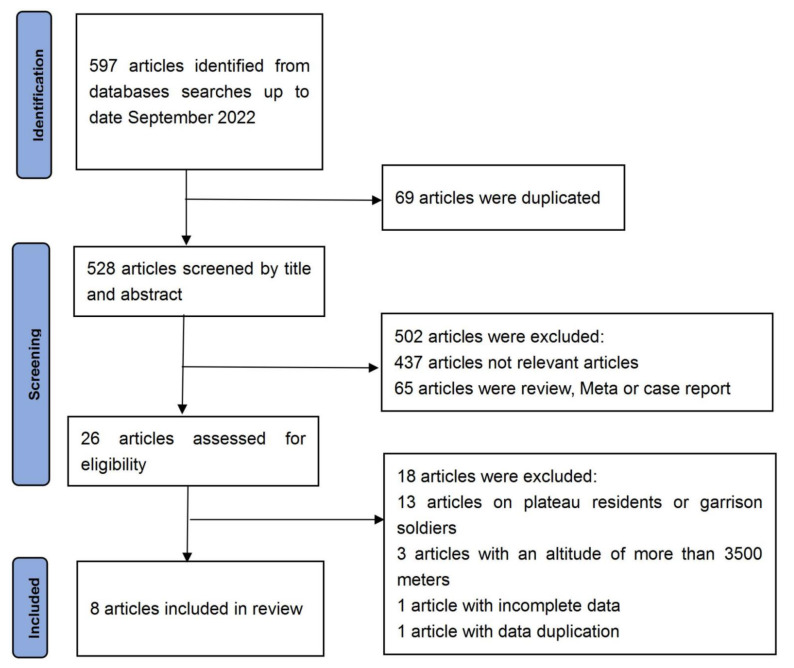
Flow chart of literature screening.

**Table 1 ijerph-20-05101-t001:** Four domains of cognitive functions consisting of eight examinations.

Domains	Examinations
Executive function	Trail-Making Test Part B (TMB)
Motor speed	Finger Tapping Test-Left (FTL) Finger Tapping Test-Right (FTR)
Memory function	Digit Span Test Forward (DSF) Digit Span Test Backward (DSB) Wechsler Memory Scale Visual (WMSV)
Verbal function	Aphasia Screening Test-Visual Motor Errors (AST-Vis) Aphasia Screening Test-Verbal Items (AST-Ver)

**Table 2 ijerph-20-05101-t002:** Eight included high-altitude mountaineering studies for cognitive function.

Studies	Type of Study	Date of Issue	Country	Age	Group Size	above Sea Level (m)	Test Period	Used Supplemental Oxygen	Indicators
Hornbein	Between Subject Design	1989	USA and Canada	24–45	41	5488	40 days	None	FTL, FTR, WMSV, AST-Vis, AST-Ver
Charles	Within-Subject Design	1983	USA	31.1 ± 8.5	22	8848	20 weeks	Above 7315 m	TMB, DSF, AST-vis
Gregor	Within-Subject Design	1989	Canada	26–49	9	8848	3–10 weeks	Above 7500 m	FTL, FTR
YouAn Wu	Within-Subject Design	1994	China	18.7	58	3680	8–10 days	--	DSF
Philip	Within-Subject Design	1995	USA	35–52	5	8848	2 months	Above 8000 m	DSF, DSB
Carole	Within-Subject Design	1988	USA	33.8 ± 3.8	8	7719	37 days	--	TMB, FTL, FTR, DSF, WMSV, AST-Ver
Gregory	Within-Subject Design	2009	Australia	Male: 34.9, Female: 32.5	26	5100	18 days	--	TMB, DSF, DSB
Carine	Within-Subject Design	2016	France	29.2 ± 1.6	4	8043	6 weeks	None	DSF, DSB

Abbreviations: TMB = Trail-Making Test Part B; FTL = Finger Tapping Test-Left; FTR = Finger Tapping Test-Right; DSF = Digit Span Test-Forward; DSB = Digit Span Test-Backward; WMSV = Wechsler Memory Scale Visual; AST-Vis = Aphasia Screening Test-Visual Motor Errors; AST-Ver = Aphasia Screening Test-Verbal Items.

**Table 3 ijerph-20-05101-t003:** Effective size (ES) and forest plot of TMB.

Effect Size					Forest Plot
Studies	Weight	Random, 95% CI			Random, 95% CI
		ES	Low	High	
Charles	38.31%	0.64	0.04	1.25	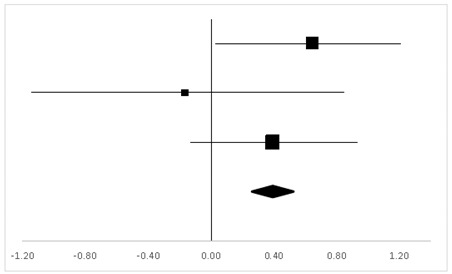
Carole	17.67%	−0.17	−1.15	0.81	
Gregory	44.02%	0.39	−0.16	0.94	
Total	100%	0.39	−0.05	0.83	Test for overall effect: Z = 1.72 (*p* > 0.05)
Heterogeneity: Q = 1.98, df = 2, C = 17.55, T^2^ = −0.001					

**Table 4 ijerph-20-05101-t004:** ES and forest plot of DSF.

Effect Size					Forest Plot
Studies	Weight	Random, 95% CI			Random, 95% CI
		ES	Low	High	
Charles	22.2%	−0.19	−0.79	0.40	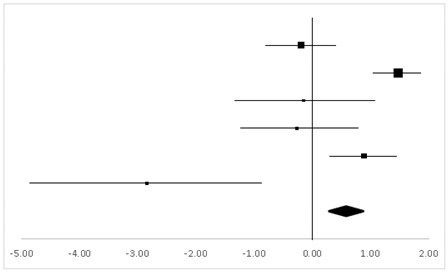
Wu	35.21%	1.47	1.06	1.88	
Philip	6.52%	−0.15	−1.39	1.09	
Carole	9.87%	−0.27	−1.25	0.72	
Gregory	23.47%	0.88	0.31	1.45	
Carine	2.74%	−2.85	−4.81	−0.88	
Total	100%	0.57	0.24	0.90	Test for overall effect: Z = 3.36 (*p* < 0.01)
Heterogeneity: Q = 42.10, df = 5, C = 40.30, T^2^ = 0.92					

**Table 5 ijerph-20-05101-t005:** ES and forest plot of DSB.

Effect Size					Forest Plot
Studies	Weight	Random, 95% CI			Random, 95% CI
		ES	Low	High	
Philip	14.92%	−1.54	−2.95	−0.13	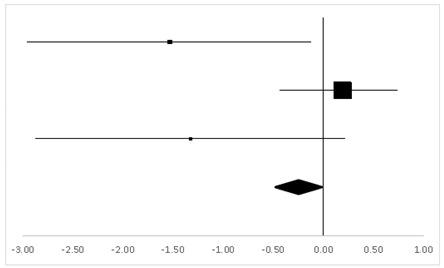
Gregory	72.28%	0.19	−0.36	0.73	
Carine	12.8%	−1.33	−2.86	0.21	
Total	100%	−0.26	−0.83	0.30	Test for overall effect: Z = −0.91 (*p* > 0.05)
Heterogeneity: Q = 7.29, df = 2, C = 7.34, T^2^ = −0.72					

**Table 6 ijerph-20-05101-t006:** ES and forest plot of FTL.

Effect Size					Forest Plot
Studies	Weight	Random, 95% CI			Random, 95% CI
		ES	Low	High	
Gregor	18.9%	0.07	−0.86	0.99	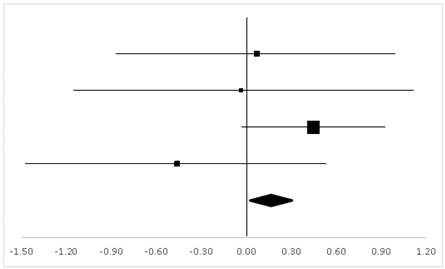
Hornbein (Operation)	13.24%	−0.04	−1.17	1.09	
Hornbein (Mountaineers)	51.18%	0.45	−0.03	0.92	
Carole	16.69%	−0.46	−1.46	0.53	
Total	100%	0.16	−0.27	0.59	Test for overall effect: Z = 0.72 (*p* > 0.05)
Heterogeneity: Q = 3.07, df = 3, C = 18.39, T^2^ = 0.004					

**Table 7 ijerph-20-05101-t007:** ES and forest plot of FTR.

Effect Size					Forest Plot
Studies	Weight	Random, 95% CI			Random, 95% CI
		ES	Low	High	
Gregor	19.4%	−0.26	−1.19	0.67	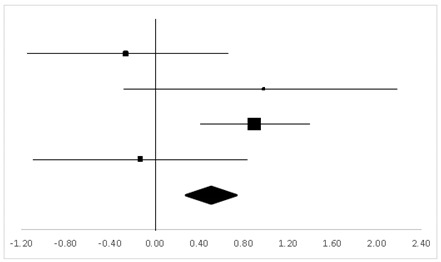
Hornbein (Operation)	12.35%	0.98	−0.22	2.18	
Hornbein (Mountaineers)	50.63%	0.89	0.4	1.39	
Carole	17.62%	−0.13	−1.11	0.85	
Total	100%	0.50	0.06	0.94	Test for overall effect: Z = −2.22 (*p* < 0.05)
Heterogeneity: Q = 7.38, df = 3, C = 17.73, T^2^ = −0.25					

**Table 8 ijerph-20-05101-t008:** ES and forest plot of the WMSV.

Effect Size					Forest Plot
Studies	Weight	Random, 95% CI			Random, 95% CI
		ES	Low	High	
Hornbein (Operation)	14.51%	1.18	−0.04	2.41	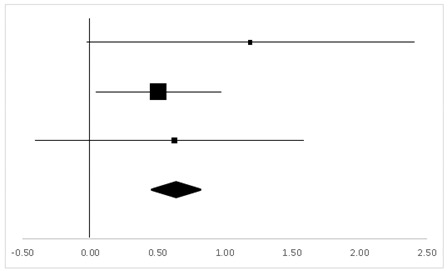
Hornbein (Mountaineers)	64.69%	0.51	0.03	0.98	
Carole	20.8%	0.62	−0.38	1.63	
Total	100%	0.63	0.14	1.12	Test for overall effect: Z = 2.52 (*p* < 0.05)
Heterogeneity: Q = 0.9, df = 2, C = 11.28, T^2^ = −0.1					

**Table 9 ijerph-20-05101-t009:** ES and forest plot of AST-Ver.

Effect Size					Forest Plot
Studies	Weight	Random, 95% CI			Random, 95% CI
		ES	Low	High	
Hornbein (Operation)	16.1%	−0.38	−1.52	0.76	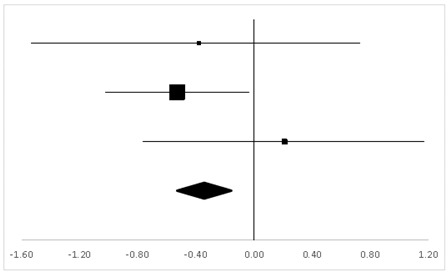
Hornbein (Mountaineers)	62.88%	−0.53	−1.01	−0.05	
Carole	21.01%	0.21	−0.77	1.19	
Total	100%	−0.35	–0.83	0.13	Test for overall effect: Z = −1.43 (*p* > 0.05)
Heterogeneity: Q = 1.84, df = 2, C = 12.07, T^2^ = −0.01					

**Table 10 ijerph-20-05101-t010:** ES and forest plot of AST-Vis.

Effect Size					Forest Plot
Studies	Weight	Random, 95% CI			Random, 95% CI
		ES	Low	High	
Charles	37.22%	0.27	−0.33	0.86	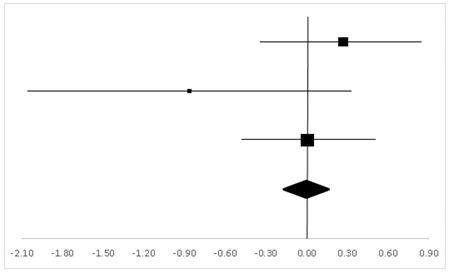
Hornbein (Operation)	11.94%	−0.87	−2.05	0.32	
Hornbein (Mountaineers)	50.84%	0	−0.47	0.47	
Total	100%	0	−0.43	0.42	Test for overall effect: Z = −0.02 (*p* > 0.05)
Heterogeneity: Q = 2.83, df = 2, C = 18.28, T^2^ = 0.05					

## Data Availability

The data that support the findings of this study are available from the first author, upon reasonable request.

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
