# Peer review of "The Effects of High-Altitude Mountaineering on Cognitive Function in Mountaineers: A Meta-Analysis"

_ijerph, 2023, doi:10.3390/ijerph20065101_

Round 1
Reviewer 1 Report
The aim of the text is to provide information about the possible impairment on cognitive functions in high altitude mountaineering. This subject is of high interest amoung the mountaineering community and also for rescue teams working at high altitude. Regarding this and the impact on safety of this issue, it is mandatory that the information given is clear and cannot lead to misundertanding.
There are few aspects I find important to highlight: The therm "anoxia" is used as a synonimus of "hypoxia" in the text, which is not correct from my point of view. On the other hand, I cannot see the information about when are the tests performed, regarding the time elapsed between the maximal exposure highlighted in Table 2 and the cognitive tests. I guess tests weren't performed at the summit of the mountains of the table. I would like to have this information as a reader, as the information can be highly confusing if the alpinist has climbed to an 8000 peak but measurements are done at 6000 meters high, and cognitive function is very different from one point to the other.
Taking into consideration that the main practical application of this paper would be improving safety choices and decisions regaring decision taking, how data is given in the text can be highly confusing and lead to safety problems if assuming the conclusions that are shown. The same for the use of oxygen, ¿Where this measurements made with the use of oxygen? ¿At which flow so, at which equivalent altitude did that correspond? This is a major issue, and although it is highlighted as a limitation of the analysis, the discussion and the table is incomplete without this information as physiologycal stress is not comparable.
Reviewer 2 Report
Problem well defined and clear. The state of the art is complete and well structured, validating the problem and the reasons for the pertinence of the study. The objectives of the study are also well defined and clear.
The description of the method used is very detailed and complete, the search strategy was well designed and its description is clear, the selection of the study is not very clear, it is not clear how it went from 597 articles in the first selection to 528 in a second selection and later only 26 articles were read. This filtering from 528 articles to 26 needs to be better explained and described in the text. After seeing figure 1 I understood the method and the filtering but in the text this action is not perceptible and makes the article confusing.
In figure 1, I recommend that the text be placed vertically in the column with the blue color, in the format in which it is, it’s reading is confusing and even letters are missing in some words. This situation must be corrected.
The discussion of the results obtained guarantees useful information for all stakeholders related to climbers and/or who in any way study physical and cognitive effects related to altitude.
Round 2
Reviewer 1 Report
Changes have improved the quality of the paper. Thank you